# Serological Response to COVID-19 Vaccination in Saudi Arabia: A Comparative Study of IgG and Neutralising Antibodies Across Vaccine Platforms

**DOI:** 10.3390/vaccines13101042

**Published:** 2025-10-10

**Authors:** Mariam M. AlEissa, Ahdab A. Alsaieedi, Reema Alduaiji, Fahad Almsned, Yousif AlDossary, Nada Saleh, Raghad A. AlQurashi, Esraa A. Hawsa, Muath b Ben Shaded, Amer M. Alshehri, Osamah T. Khojah, Eyad Y. Abu Sarhan, Hamad H. Alonazi, Walid A. Nouh, Khalid H. AlAnazi, Sami S. Almudrra, Khaled I. AlAbdulkareem, Abdullah AlJurayyan, Abdullah M. Asiri

**Affiliations:** 1College of Medicine, Alfaisal University, Riyadh 11533, Saudi Arabia; abdullahm.asiri@moh.gov.sa; 2Ministry of Health, Riyadh 11176, Saudi Arabia; 3Public Health Lab, Public Health Authority, Riyadh 11176, Saudi Arabia; 4King Khaled Eye Specialist Hospital (KKESH) Research Center, Riyadh 11462, Saudi Arabia; 5Computational Sciences Department, Centre for Genomic Medicine (CGM), King Faisal Specialist Hospital & Research Center, Riyadh 11211, Saudi Arabia; 6Department of Medical Laboratory Sciences, Faculty of Applied Medical Sciences, King Abdulaziz University, Jeddah 21589, Saudi Arabia; 7King Fahd Medical Research Center, King Abdulaziz University, Jeddah 21589, Saudi Arabia; 8John Hopkins Aramco Healthcare, ARAMCO Saudi Arabia, Dhahran 31311, Saudi Arabia; 9Research Center, King Fahad Specialist Hospital in Dammam (KFSH-D), Dammam 32253, Saudi Arabia; 10Research Program, Academic, Training, and Research Administration, Eastern Health Cluster, Dammam 32253, Saudi Arabia; 11School of Systems Biology, George Mason University, Fairfax, VA 22030, USA; 12College of Medicine, King Saud bin Abdulaziz University for Health Sciences (KSAU-HS), Riyadh 11481, Saudi Arabia; 13King Abdullah International Medical Research Centre (KAIMRC), Ministry of National Guard Health Affairs (MNGHA), Riyadh 11426, Saudi Arabia; 14Pathology Department, College of Medicine, King Saud University, Riyadh 12372, Saudi Arabia; 15Medical Diagnostic Laboratories, Sulaiman Alhabib Medical Group, Riyadh 12511, Saudi Arabia; 16Gulf Center for Disease Prevention and Control, Gulf Health Council, Riyadh 11481, Saudi Arabia; 17Immunology and Serology Laboratory, Pathology and Clinical Laboratory Administration, King Fahad Medical City, Riyadh 11525, Saudi Arabia

**Keywords:** SARS-CoV-2, COVID-19, vaccine immunogenicity, neutralising antibodies, Saudi Arabia, mRNA vaccine, serological response

## Abstract

**Background**: In the Kingdom of Saudi Arabia, various COVID-19 vaccines were administered during the pandemic. However, region-specific real-word comparative data on their immunogenicity remain limited. This study aimed to assess the serological responses to Pfizer-BioNTech (BNT162b2), Moderna (mRNA-1273), and AstraZeneca (ChAdOx1 nCoV-19) vaccines in a diverse population living in KSA. **Methods**: This observational study included 236 adults recruited from vaccination sites in Riyadh. Participants provided serum samples at predefined intervals: before the first dose, after the first dose, after the second dose, and post-vaccination infection (if applicable). IgG and neutralising antibodies were quantified using ELISA assays. Demographic and vaccination data, and their associations with antibody responses, were evaluated. **Results**: At baseline, 75.4% of participants were positive for SARS-CoV-2 IgG, suggesting high prior exposure. Marked incremental increases in IgG levels were observed after each vaccine dose. Both Moderna and Pfizer elicited stronger responses, with Pfizer inducing the strongest early response and Moderna achieving the highest overall titres. Among IgG-positive individuals, neutralising antibodies were detected in 98.1%. There were no statistically significant differences by age or gender, although males tended to show higher mean titres. Heterologous vaccine schedules induced comparable or enhanced immunogenicity relative to homologous schedules, supporting their use in flexible immunisation strategies. **Conclusions**: All COVID-19 vaccines administered in Saudi Arabia elicited robust antibody responses, particularly the mRNA-based vaccines. Our findings support their continued use and justify varied vaccination approaches, including mix-and-match booster strategies, to enhance community immunity.

## 1. Introduction

An unknown respiratory disease was documented in December 2019 in Wuhan, the province of Hubei, China [1]. Next-generation sequence (NGS) was performed on samples obtained from infected patients, and the results identified the etiological agent as a novel betacoronavirus [2]. Scientists were subsequently able to isolate and characterise the pathogen, now known as Severe Acute Respiratory Syndrome coronavirus 2 (SARS-CoV-2) [3]. The World Health Organisation (WHO) named the diseases coronavirus disease 2019 (COVID-19) [4,5].

The U.S. Food and Drug Administration (FDA) issued an Emergency Use Authorisation (EUA) for the Pfizer-BioNTech COVID-19 vaccine [6]. The Saudi FDA registered the vaccine and approved its importation and use in the Kingdom of Saudi Arabia [7]. The vaccine is indicated for active immunisation to prevent COVID-19 in individuals 12 years of age and older. Its mRNA, encapsulated in lipid nanoparticles, enables delivery into host cells for expression of SARS-CoV-2 spike (S) antigen, thereby triggering an immune response [6]. Subsequently, the European Medicines Agency granted conditional marketing authorisation for the AstraZeneca ChAdOx1 nCoV-19 (AZD1222) vaccine [8], which was also approved for importation and use by the Saudi FDA [9]. Since then, multiple vaccines have been developed and released, with updated regulations for age-specific eligibility and booster dose recommendations [10]. Several studies worldwide have assessed vaccine efficacy [11,12], including some research conducted in Saudi Arabia. However, our study aims to evaluate the immunological profile and validate the efficacy of COVID-19 vaccines (BNT162b2 Pfizer-BioNTech, and ChAdOx1 nCoV-19 (AZD1222) AstraZeneca) in the Saudi population.

## 2. Methodology

### 2.1. Ethical Approval

The ethical approval was obtained from the Central Institutional Review Board at the Ministry of Health Ethics Committee, Saudi Arabia (approval number IRB-21-55M; National Registry Number NCBE-KACST, KSA(H-01-r-009)). The SFDA approved the application (application number 21061802).

### 2.2. Participant Recruitment and Sample Collection

The samples were collected from participants visiting vaccination sites in the Riyadh region between October 2021 and July 2022. Recruitment was conducted collaboratively between the Public Health Authority (PHA) and the Ministry of Health (MOH). The study included Saudis and non-Saudis, healthy males and females aged 16 years and older, who were vaccinated with one of the following vaccines: Moderna (mRNA-1273) from ModernaTX, Inc., (Cambridge, MA, USA), Johnson & Johnson (JNJ-78436735) from Janssen Biotech, Inc. (a Johnson & Johnson company), (Horsham, PA, USA), Pfizer BioNTech (BNT162b2)from Pfizer Inc., (New York, NY, USA), and BioNTech SE, Mainz, Germany, or AstraZeneca (ChAdOx1 nCoV-19) from AstraZeneca, (Cambridge, United Kingdom). Pregnant women and individuals with prior viral infection were excluded. Samples were collected at three time points: baseline, post-dose 1, and post-dose 2. A fourth group included participants with confirmed COVID-19 infection despite vaccination. All participants provided informed consent to participate in this study and for follow-up sample collection. A sample of 4–5 mL of blood was collected in a plain tube from each participant and left for 30 min to clot. Then, the samples were transported to the PHA laboratory at 2–8 °C and received on the same day. Each participant was assigned an identifier number. The samples were subsequently centrifuged at 1000–1300× *g* for 10 min.

### 2.3. IgG & Neutralising Antibody Assays

The serum was isolated in a separate tube and stored at −80 °C. IgG levels were measured using Abbot’s SARS-CoV-2 IgG enzyme-linked immunosorbent assay (ELISA) kit (Abbott Laboratories, Abbott Park, IL, USA). The neutralisation assay on serum samples was conducted using the Invitrogen competitive SARS-CoV-2 Neutralising Antibody ELISA Kit, following the manufacturer’s instructions. Prior to the assay, the serum samples were diluted 1:50 with 1× Assay Buffer. The neutralisation percentage (%) was determined using the formula:
Neutralisation (%) = [1 − (absorbance of the unknown sample/absorbance of the negative control)] × 100

The methodology, as described previously by Bhadauria et al. and Choe et al. [13,14], involved reporting the mean inhibition percentages observed in duplicate wells. To determine the presence of neutralising antibodies against the SARS-CoV-2-RBD antigen, a positivity cutoff of ≥20.0% was established. Percentages below 20% were considered negative, based on the set criteria [13,14].

### 2.4. Statistical Analysis

Statistical analyses were conducted using IBM SPSS Statistics version 30.0. This was a longitudinal study with a high dropout rate, which resulted in substantial missing data; therefore, imputation techniques were considered inappropriate. Cases with only a single data point were excluded. The remaining cases were classified into two analytical datasets: a paired and an unpaired dataset.

The paired dataset included participants with complete IgG recordings across all three time points and was analysed to assess within- and between-subject effects using repeated-measures ANOVA. The unpaired dataset comprised both the paired participants and those with one missing time point, with each data point treated as a separate observation. This approach provided a confirmatory analysis of the between-group differences identified in the paired dataset, using the Mann–Whitney U test and the Kruskal–Wallis test.

## 3. Results

After applying exclusion criteria and accounting for missing data, the study included a total of 236 participants, of whom only 36 (15.25%) were paired participants. The participants were 110 females (46.6%) and 126 males (53.4%). Most participants were aged 31–45 years (45.3%), followed by 18–30 years (29.7%), while only 2.1% were above 60 years. Chronic illness was reported by 19.1% of participants, most commonly hypertension (4.2%), diabetes (3.4%), asthma (1.7%), and cancer (1.7%).

The majority received the Pfizer vaccine as the first dose (83.9%), while 7.2% received AstraZeneca and 8.9% Moderna. For the second dose, Pfizer remained the most frequent (43.6%), followed by Moderna (19.9%) and AstraZeneca (2.5%), though one-third had unknown records. Only a small subset (3.4%) received heterologous schedules, while 10.2% received homologous schedules, with the remainder undocumented. At baseline, 75.4% were IgG positive, indicating previous infection with COVID-19. A total of 5.9% of the participants were infected after two doses; all of these were female (Table 1, Figure 1).

### 3.1. Overall IgG Response Across Time Points

Analysis of the paired sample showed a significant effect of collection point on IgG levels (F(2,34) = 75.36, *p* < 0.001). IgG increased sharply from baseline (M = 350 AU/mL) to post-first dose (M = 16,350 AU/mL) and post-second dose (M = 18,605 AU/mL). Pairwise comparisons revealed significant increases from baseline to both post-dose points (*p* < 0.001), with no difference between the two post-dose measures (*p* = 0.818). (Table 2 and Table 3). The findings were consistent with those of the unpaired group, as shown in Figure 2.

### 3.2. Impact of Demographic Factors (Co-Variates)

#### 3.2.1. Gender

While males’ IgG readings were higher than those of females, the analysis of the paired dataset revealed that gender did not significantly moderate changes in IgG levels over time (F(2,33) = 1.26, *p* = 0.297), nor was there a significant main effect of gender on IgG levels (F(1,34) = 2.10, *p* = 0.157). See Table 2 and Table 3. The unpaired dataset analysis was consistent with the paired dataset findings, confirming that gender was not a significant between-subject factor and that immune responses were comparable across genders, as shown in Figure 3.

#### 3.2.2. Age Groups

The results of age group analysis indicated no significant interaction between collection point and age group (F(6,60) = 0.54, *p* = 0.774), and age group did not significantly affect overall IgG levels (F(3,31) = 0.88, *p* = 0.463). See Table 2 and Table 3. The analysis of the unpaired dataset for age as a between-subject factor was consistent with the paired dataset findings., as shown in (Figure 4).

#### 3.2.3. Chronic Disease Status

Chronic illness status was also tested as a within- and between-subjects factor, and neither significantly moderated changes over time (F(2,33) = 0.10, *p* = 0.905) nor had a main effect on IgG levels (F(1,34) = 0.20, *p* = 0.654). See Table 2 and Table 3. The findings were aligned with the unpaired dataset analysis, as shown in Figure 5.

### 3.3. Vaccine Platform Comparison

The analysis for the paired group revealed no significant interaction between collection point and first-dose vaccine type (F(2,33) = 2.60, *p* = 0.090), though Pfizer recipients exhibited higher mean IgG levels compared to AstraZeneca and Moderna at both time points. For the second dose, there was a significant interaction between collection point and second-dose vaccine type (F(2,29) = 5.67, *p* = 0.008), See Table 2 and Table 3. For both first and second doses, vaccine type was not statistically significant as a between-subject factor in the paired group, as illustrated in Figure 6.

In the unpaired group, second-dose vaccine type was a significant between-subject factor (*p* = 0.030). Pairwise comparisons revealed a statistically significant difference between Moderna and AstraZeneca (*p* = 0.012), while the difference between Moderna and Pfizer approached significance but did not reach the threshold (*p* = 0.056).

### 3.4. Heterologous vs. Homologous Schedules

The heterologous and homologous schedules were tested as a between-subjects factor. There was no significant interaction between collection point and vaccination schedule (F(1,30) = 1.36, *p* = 0.252), nor a significant main effect of schedule (F(1,30) = 0.15, *p* = 0.703). This was consistent with the unpaired dataset analysis.

As shown in Figure 7, both heterologous and homologous recipients exhibited similar IgG distributions, with a trend toward higher median levels in the heterologous group at post-second dose, though differences were not significant.

We also investigated the serum neutralising activity of 162 participants who had previously tested positive for SARS-CoV-2 IgG antibodies, with a median IgG titre exceeding 6620.35 AU/mL. Among these participants, 159 of 162 (98.1%) exhibited positive neutralising activity within the two groups (Figure 8). Our findings indicated a slight, non-significant increase in serum neutralising activity in patients who were previously infected after receiving the second dose, compared with non-infected participants who had received the COVID-19 vaccination after the second dose (53.3% and 50.6%, respectively) (*p* value = 0.343).

## 4. Discussion

This study provides insight into the serological response to the Pfizer-BioNTech (BNT162b2), Moderna (mRNA-1273), and AstraZeneca (ChAdOx1 nCoV-19) vaccines administered in Saudi Arabia in 2021. Participants showed high baseline IgG positivity prior to vaccination, with 73.1% positivity, suggesting significant undocumented or asymptomatic SARS-CoV-2 exposure. This level of positivity does not result from prior exposure to other endemic human coronaviruses due to the high specificity of the detection method employed. However, these figures are considerably greater than post-vaccine levels measured in this study, indicating waning IgG levels in line with the decline observed after months of infection [15]. This reinforces the likelihood that undocumented previous infections contributed to our cohort’s relatively high seroprevalence. This baseline finding also aligns with previous seroprevalence studies conducted in Saudi Arabia, which reported exposure within the community during the early stages of the pandemic [16,17].

The significant rise in IgG levels following administration of the first dose, especially with the Pfizer vaccine, is consistent with many studies demonstrating the strong immunogenicity of mRNA platforms [18]. Pfizer’s strategy of utilising a nucleoside-modified mRNA encoding the spike protein of SARS-CoV-2, delivered via lipid nanoparticles, effectively promotes dendritic cell presentation and elicits a rapid antibody response [19]. AstraZeneca’s more conservative response to the first dose may be explained by pre-existing immunity to the adenoviral vector, which reduces the effectiveness of this vaccine [20].

Following the second dose, we noted the greatest increase in IgG levels in individuals who received the Moderna vaccine. This is consistent with similar studies of immunogenicity conducted in the U.S. and Europe, which showed that Moderna’s vaccines elicited higher peak antibody titres compared with Pfizer’s [21]. The disparities might be partially explained by the greater mRNA content of Moderna’s vaccine (100 µg vs. 30 µg for Pfizer) and the longer interval between doses (28 days vs. 21 days). Our results align with recent international reports, such as Barros-Martins et al. (2021) and Pozzetto et al. (2021) [11,22], which demonstrated that heterologous vaccine regimens, especially combinations of vector priming and mRNA boosting, appear to enhance immunogenicity. In our study, heterologous vector-mRNA schedules elicited comparable or superior IgG responses relative to homologous regimens, particularly when mRNA platforms were included. Similarly, durable IgG responses have been observed in regional studies from Saudi Arabia [23]. However, unlike those studies, our study directly assessed both binding (IgG) and functional neutralising antibodies across multiple two-dose platforms within the Saudi population. This provides important, region-specific evidence that complements the global literature and addresses the limited data available from the Middle East. These findings also contribute to the expanding evidence from other countries and reinforce the need for flexible vaccination planning in response to changing epidemiological conditions [24].

The results regarding neutralising antibodies from our study confirm the findings related to IgG, as 98.1% of those who tested positive for IgG also had neutralising activity. Such a high agreement reinforces the conclusion that anti-spike IgG titres can serve as proxies for functional protection. Many researchers have reported high IgG levels and neutralisation capacity, alongside expected protective clinical outcomes against symptomatic COVID-19 and hospitalisation [25]. This concordance also adds confidence that our serological findings are meaningful proxies of protective immunity, as highlighted in multiple reviews [25,26]. These results further strengthen WHO and local policy recommendations regarding prioritisation of second-dose completion and support the use of antibody monitoring in evaluating vaccination strategies.

In our study, we found no significant differences in IgG titres by age or gender. This was expected, given that our study population was relatively young and therefore unlikely to demonstrate age-related differences due to immunosenescence [27]. Although some literature indicates that females tend to have stronger antibody responses to vaccination [28,29], this pattern may not have been captured in our dataset due to insufficient statistical power. Further Saudi-based age-stratified studies are needed to explore the impact of these variables, particularly in elderly and immunocompromised populations.

One of the strengths of our study is the assessment of vaccine combinations and their immune responses in a Middle Eastern population, which has been underrepresented in vaccine research. This region-specific evidence strengthens the contribution of our study, since regional data on COVID-19 vaccine immunogenicity remain sparse compared with Western cohorts, yet are essential for shaping public health policies tailored to Saudi Arabia. Nonetheless, it is important to consider some limitations. Primarily, a significant part of our study population lacked follow-up data beyond the second vaccine dose. This was due to challenges in data collection during an active public health crisis, especially during rapidly evolving vaccination campaigns and shifting public health strategies. These gaps limited our ability to analyse antibody decline or medium-term responses beyond the early post-vaccination phase.

In addition, while the serological data presented here are essential for understanding levels of humoral immunity, we did not assess T-cell responses or the activation of memory B-cells. The ability to mount long-term protection, especially against newly emerging variants, depends significantly on cellular immunity. With the continuous emergence of new variants, there is a need for future longitudinal research focused on vaccine efficacy that integrates cellular immunity data.

As for other limitations, although neutralising antibodies were detected using an ELISA-based competition inhibition method, we did not conduct live-virus or pseudovirus neutralisation assays, which could shed light on variant-specific protection. These approaches are especially important in light of newer subvariants such as Omicron BA.5, XBB, and others that test the limits of vaccine-induced immunity [30]. This limitation is particularly relevant given published evidence that first-generation Wu-1-based vaccines elicit little or no neutralising activity against newer Omicron NB sublineages, underscoring the importance of updated vaccine formulations [29,30,31]. Recent studies have shown that bivalent and updated mRNA vaccines targeting Omicron sublineages elicit broader neutralisation profiles, reflecting the continual evolution of vaccine strategies in response to emerging variants [31,32].

Regardless of the constraints, our study has important implications for public health policy in Saudi Arabia and comparable regions. First, it confirms that all licensed COVID-19 vaccines elicit robust immune responses, particularly when full vaccination courses are completed. Second, the finding that Moderna’s mRNA vaccine achieved higher efficacy further supports its prioritisation in booster campaigns among high-risk groups or its long-term use in vulnerable populations. Lastly, the demonstrated potential for a platform-agnostic approach could help guide national policy strategies, especially during vaccine shortages.

Nonetheless, sustained surveillance of population-level antibody dynamics can help track community/herd immunity thresholds and inform strategic preparedness for pandemics. Surveillance is particularly valuable for distinguishing between immunity induced by vaccine and infection in populations such as Saudi Arabia’s, where mixed immunity patterns exist. Our results should encourage public health officials in Saudi Arabia to incorporate immune response data into vaccine policy planning. This further highlights the importance of developing national immunological surveillance programs, as emphasised in Saudi Vision 2030 health security initiatives.

## 5. Conclusion

The response to COVID-19 vaccination in Saudi Arabia showed strong IgG and neutralising antibody responses, particularly with mRNA vaccines. Pfizer generated the strongest early IgG response, while Moderna achieved the highest titres after the second dose. Most participants with positive IgG also demonstrated neutralising activity. These findings support ongoing vaccine deployment and the adoption of flexible vaccination strategies, such as heterologous boosting, to enhance population immunity. Taken together, our results not only confirm global trends but also provide valuable regional-specific evidence that can inform Saudi vaccination policy.

## Figures and Tables

**Figure 1 vaccines-13-01042-f001:**
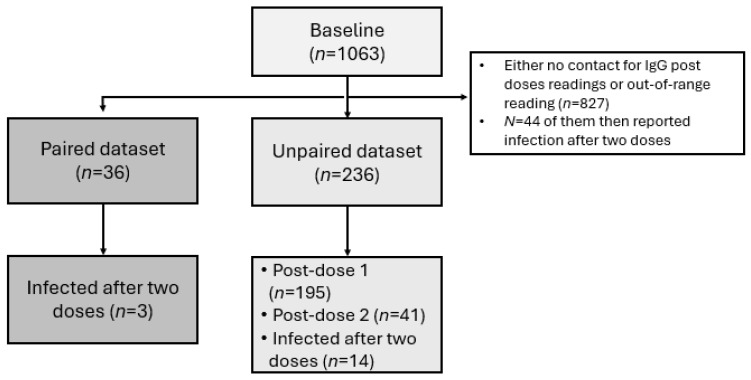
Flow chart of study participants.

**Figure 2 vaccines-13-01042-f002:**
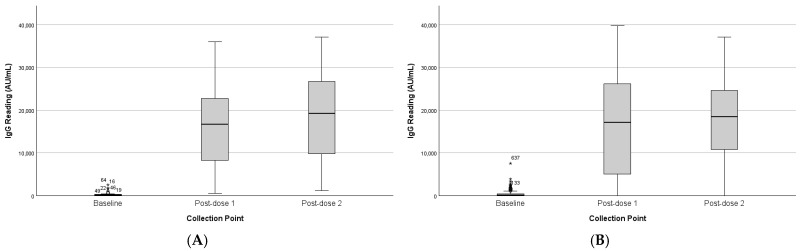
IgG levels across collection points. (**A**) Paired dataset (*n* = 36). (**B**) Unpaired dataset (*n* = 236). (*) represent outliners. Pairwise comparisons showed significant differences between baseline vs. post-dose 1 and baseline vs. post-dose 2 (both *p* < 0.001).

**Figure 3 vaccines-13-01042-f003:**
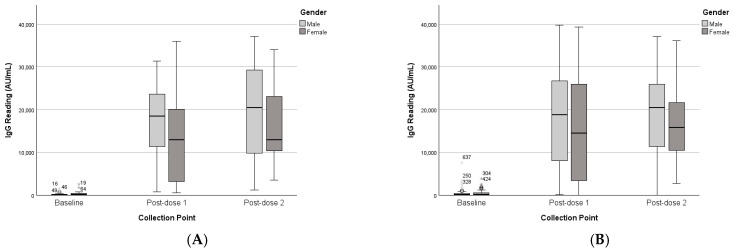
IgG levels of paired and unpaired groups across collection points and gender. (**A**) Paired dataset. (**B**) Unpaired dataset. (*) represent outliners.

**Figure 4 vaccines-13-01042-f004:**
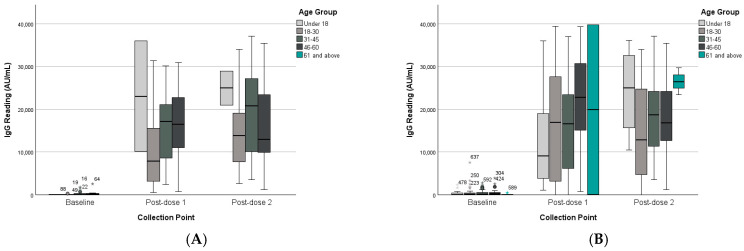
IgG levels of paired and unpaired groups across collection points and age groups. (**A**) Paired dataset. (**B**) Unpaired dataset. (*) represent outliners.

**Figure 5 vaccines-13-01042-f005:**
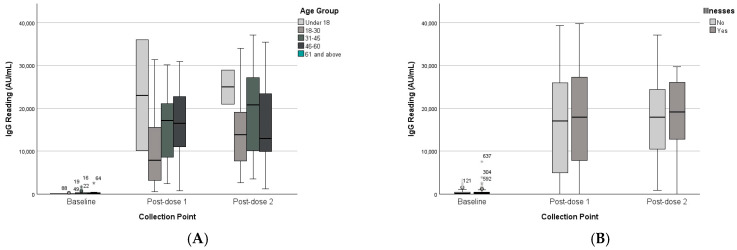
IgG levels of paired and unpaired groups across collection points and chronic disease status. (**A**) Paired dataset. (**B**) Unpaired dataset. (*) represent outliners.

**Figure 6 vaccines-13-01042-f006:**
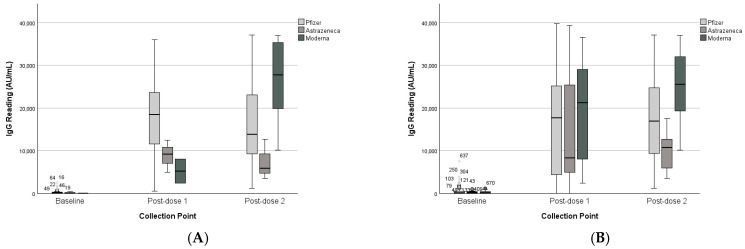
IgG levels of paired and unpaired groups across collection points and vaccine type received before the reading (baseline did not receive vaccine, therefore classified based on the first dose). (**A**) Paired dataset. (**B**) Unpaired dataset. (*) represent outliners. Pairwise comparisons showed significant differences between Moderna vs. AstraZeneca (*p* < 0.012).

**Figure 7 vaccines-13-01042-f007:**
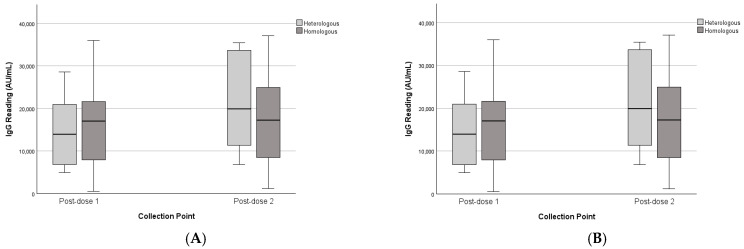
IgG levels of paired and unpaired groups across collection points and heterologous vs. homologous schedules. (**A**) Paired dataset. (**B**) Unpaired dataset.

**Figure 8 vaccines-13-01042-f008:**
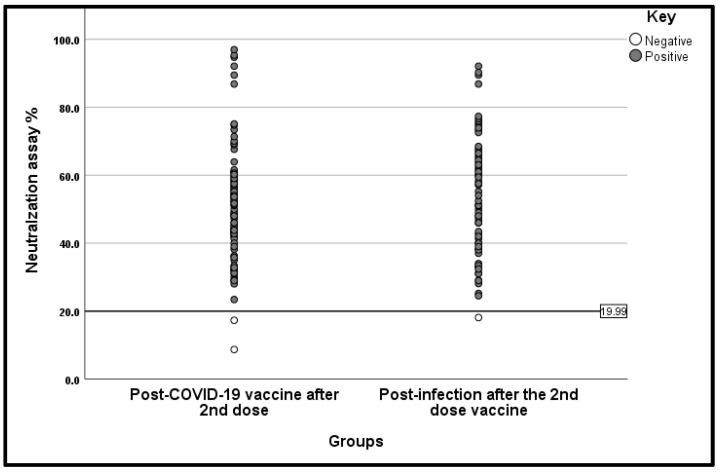
Neutralisation Assay result distributed by patient post-COVID-19 vaccination.

**Table 1 vaccines-13-01042-t001:** Participant Characteristics.

	Female (*n* = 110, 46.6%)	Male (*n* = 126, 53.4%)	Total (*n* = 236, 100%)
Age group			
Under 18	9 (3.8%)	6 (3%)	15 (6.3%)
18–30	35 (14.8%)	35 (14.8%)	70(29.7%)
31–45	49 (20.8%)	58 (24.6%)	107 (45.3%)
46–60	15 (6.4%)	18 (7.6%)	33 (14%)
61 and above	0 (0%)	5 (2.1%)	5 (2.1%)
Unknown	2 (0.8%)	4 (1.7%)	6 (2.5%)
Does the participant have a chronic illness?			
Yes	21 (8.9%)	24 (10.2%)	45 (19.1%)
No	89 (37.7%)	102 (43.2%)	191 (80.9%)
Type of chronic illness			
Dementia	0 (0.0%)	0 (0.0%)	0 (0.0%)
Allergy	2 (0.8%)	1 (0.4%)	3 (1.3%)
Blood Disorders	2 (0.8%)	0 (0%)	2 (0.8%)
Diabetes	2 (0.8%)	6 (2.5%)	8 (3.4%)
HTN	3 (1.3%)	7 (3%)	10 (4.2%)
HIT	1 (0.4%)	0 (0%)	1 (0.4%)
Cholesterol	0 (0%)	4 (1.7%)	4 (1.7%)
Stroke	0 (0.0%)	0 (0.0%)	0 (0.0%)
Cancer	3 (1.3%)	1 (0.4%)	4 (1.7%)
Renal disease	0 (0%)	3 (1.3%)	3 (1.3%)
Heart problems	0 (0.0%)	0 (0.0%)	0 (0.0%)
Rhesus disease	0 (0.0%)	0 (0.0%)	0 (0.0%)
Respiratory failure	0 (0.0%)	0 (0.0%)	0 (0.0%)
Asthma	2 (0.8%)	2 (0.8%)	4 (1.7%)
Other	11 (4.7%)	7 (3%)	18 (7.6%)
Type of Vaccine in the First Dose			
Pfizer	97 (41.1%)	101 (42.8%)	198 (83.9%)
AstraZeneca	8 (3.4%)	9 (3.8%)	17 (7.2%)
Moderna	5 (2.1%)	16 (6.8%)	21 (8.9%)
Type of Vaccine in the Second Dose			
Pfizer	48 (20.3%)	55 (23.3%)	103 (43.6%)
AstraZeneca	2 (0.8%)	4 (1.7%)	6 (2.5%)
Moderna	17 (7.2%)	30 (12.7%)	47 (19.9%)
Unknown	43 (18.2%)	37 (15.7%)	80 (33.9%)
Heterologous vs. homologous schedules			
Heterologous	2 (0.8%)	6 (2.5%)	8 (3.4%)
Homologous	10 (4.2%)	14 (5.9%)	24 (10.2%)
Unknown	98 (41.5%)	106 (44.9%)	204 (86.4%)
Baseline (i.e., before 1st dose)			
Positive	80 (33.9%)	98 (41.5%)	178 (75.4%)
Negative	30 (12.7%)	28 (11.9%)	58 (24.6%)
Post-dose 1 (i.e., after 20 days from 1st dose)			
Positive	93 (39.4%)	102 (43.2%)	195 (82.6%)
Negative	0 (0.0%)	0 (0.0%)	0 (0.0%)
Unknown	17 (7.2%)	24 (10.2%)	41 (17.4%)
Post-dose 2 (i.e., after 20 days from 2nd dose)			
Positive	30 (12.7%)	47 (19.9%)	77 (32.6%)
Negative	0 (0.0%)	0 (0.0%)	0 (0.0%)
Unknown	80 (33.9%)	79 (33.5%)	159 (67.4%)

**Table 2 vaccines-13-01042-t002:** Descriptive statistics for IgG levels in paired and unpaired datasets.

		Paired Sample						Unpaired Sample					
		Baseline		Post-Dose 1		Post-Dose 2		Baseline		Post-Dose 1		Post-Dose 2	
Variable	Values	Mean	SD	Mean	SD	Mean	SD	Mean	SD	Mean	SD	Mean	SD
Collection Point	Baseline	350	567	-	-	-	-	435	757	-	-	-	-
	Post-dose 1	-	-	16,350	9827	-	-	-	-	17,019	12,009	-	-
	Post-dose 2	-	-	-	-	18,605	10,680	-	-	-	-	18,140	9617
Gender	Male	276	388	18,001	8800	19,994	11,490	429	838	18,206	11,363	19,163	10,099
	Female	480	796	13,429	11,190	16,146	8967	443	657	15,717	12,613	16,539	8732
Age Group	Under 18	49	55	23,073	18,296	24,966	5673	382	713	13,707	12,482	24,141	11,000
	18–30	94	120	11,734	12,366	15,467	12,105	459	1039	16,848	13,302	14,534	10,930
	31–45	394	571	16,393	8625	20,167	11,088	418	535	16,037	11,027	18,595	9324
	46–60	417	724	16,479	9950	16,162	10,764	548	813	21,664	11,314	17,292	9088
	61 and above	-	-	-	-	-	-	137	194	19,908	28,018	26,504	3132
	Unknown	745	0	23,837	0	21,882	0	242	265	21,898	6021	21,882	0
Chronic Illness	No	302	460	15,968	10,600	18,468	11,438	379	546	16,757	11,994	18,176	9699
	Yes	587	971	18,259	4366	19,289	6241	674	1305	18,177	12,178	17,994	9596
Type of vaccine (post-dose 1)	Pfizer	386	601	17,788	9761	-	-	460	813	16,885	11,850	-	-
	AstraZeneca	204	186	8916	3761	-	-	328	311	14,589	14,079	-	-
	Moderna	7	10	5217	3986	-	-	286	341	20,126	11,864	-	-
Type of vaccine (post-dose 2)	Pfizer	-	-	17,077	10,527	16,780	10,493	-	-	16,901	11,878	17,381	9942
	AstraZeneca	-	-	12,139	2764	7365	4742	-	-	10,950	3279	10,088	5554
	Moderna	-	-	13,471	9294	26,657	9715			19,143	12,626	25,426	8475
	Unknown	515	740	21,450	10,150	20,506	5835	-	-	15,967	12,078	17,059	8729
Heterologous vs. homologous	Heterologous	-	-	14,630	8575	21,516	11,471	-	-	14,630	8575	21,516	11,471
	Homologous	-	-	16,073	10,272	17,317	11,125	-	-	16,073	10,272	17,317	11,125
	Unknown	-	-	21,450	10,150	20,506	5835	-	-	17,276	12,412	17,979	8464

**Table 3 vaccines-13-01042-t003:** Repeated-measures ANOVA—paired sample.

	F	df	Error df	*p*-Value
Collection Point	75	2	34	**<0.001 ****
Gender	2.10	1	34	0.157
Gender × Collection Point	1.26	2	33	0.297
Age Group	0.88	3	31	0.463
Age Group × Collection Point	0.54	6	60	0.774
Chronic Illness	0.20	1	34	0.654
Chronic Illness × Collection Point	0.10	2	33	0.905
Type of vaccine (post-dose 1)	2.8	2	33	0.074
Type of vaccine (post-dose 1) × Collection Point	2.60	2	33	0.090
Type of vaccine (post-dose 2)	1.62	2	29	0.215
Type of vaccine (post-dose 2) × Collection Point	5.67	2	29	**0.008 ****
Heterologous vs. homologous	0.15	1	30	0.703
Heterologous vs. homologous × Collection Point	1.36	1	30	0.252

** *p* < 0.05.

## Data Availability

Data is available upon request by contacting the PI.

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
