# Peer review of "Serological Response to COVID-19 Vaccination in Saudi Arabia: A Comparative Study of IgG and Neutralising Antibodies Across Vaccine Platforms"

_vaccines, 2025, doi:10.3390/vaccines13101042_

Round 1
Reviewer 1 Report
Comments and Suggestions for Authors
Authors have done a fairly good job, although they did not obtain any interesting or unexpected results
The reviewer's comments are not related to the study design, but to the presentation of the results
1) In Table 1, the Unknown (sex) column looks strange, perhaps these patients should be excluded from consideration in the article?
2) In Table 2, please, round the values ​​to integers in the Denominator column (and possibly F)
3) In Tables 3, 4, 5, 6, please, round Mean and SD to integers
4) It seems that in the meaning of Fig. 4, Cocktail/Same should be replaced with Homologous/Heterologous
5) In section 2.4, please, describe in more detail which statistical tools and criteria were used and why, whether distributions were checked. The text of the article mentions ANOVA, this should be described in the Methodology section.
6) Why is there no assessment of the significance of differences between groups anywhere in the results are presented in the manuscirpt?
Author Response
Review Comments and Responses for Submission ID vaccines-3867218
We thank the reviewers and the editor for their valuable comments and suggestions provided. We have revised the manuscript accordingly, which we believe has improved the clarity, presentation and overall quality of the work. Our detailed responses to each point are provided below
Reviewer1:
The reviewer's comments are not related to the study design, but to the presentation of the results
Response: We thank the reviewers for their constructive feedback. Upon further review, we have substantially improved the clarity and consistency of the data presentation throughout the manuscript. These revisions include formatting adjustments, rounding of values and the exclusion of unknown values. The refinements were made to enhance the readability and transparency of the results. Importantly, these changes concern only data presentation; the main findings and study conclusions remain unchanged.
- In Table 1, the Unknown (sex) column looks strange, perhaps these patients should be excluded from consideration in the article?
Response: Thank you for the comment. In the revised version of the manuscript, participants with unknown gender have been excluded from Table 1 and from the analysis.
- In Table 2, please, round the values ​​to integers in the Denominator column (and possibly F)
Response: Thank you for the comment. The data analysis was revisited, and the findings are now presented in a clear and consistent manner.
- In Tables 3, 4, 5, 6, please, round Mean and SD to integers
Response: Thank you for the comment. The Mean and SD values were revisited and reported in a clear and consistent manner.
- It seems that in the meaning of Fig. 4, Cocktail/Same should be replaced with Homologous/Heterologous
Response: Thank you for the observation. We have replaced the label “Cocktail/Same” with “Homologous/Heterologous” throughout the manuscript to reflect the intended meaning.
- In section 2.4, please, describe in more detail which statistical tools and criteria were used and why, whether distributions were checked. The text of the article mentions ANOVA, this should be described in the Methodology section.
Response: Thank you for the observation. We have amended Section 2.4 (Statistical Analysis) in the Methodology to provide more details as requested. The revised text now reads as follows:
“The dataset was cleaned and prepared in Microsoft Excel to ensure accuracy and consistency, and statistical analyses were conducted using IBM SPSS Statistics version 30.0. This was a longitudinal study with a high dropout rate, which resulted in substantial missing data; imputation techniques were considered inappropriate. Cases with only a single data point were excluded. The remaining cases were classified into two analytical datasets: a paired and an unpaired dataset.
The paired dataset included participants with complete IgG recordings across all three time points and was analysed to assess within- and between-subject effects using repeated-measures ANOVA. The unpaired dataset comprised both the paired participants and those with one missing time point, with each data point treated as a separate observation. This approach provided a confirmatory analysis of the between-group differences identified in the paired dataset, using Mann–Whitney U and Kruskal–Wallis tests.”
- Why is there no assessment of the significance of differences between groups anywhere in the results are presented in the manuscript?
Response: Thank you for the observation. Pairwise comparisons were performed, but only statistically significant results were reported in the manuscript, as most comparisons did not reach significance.

Reviewer 2 Report
Comments and Suggestions for Authors
Comment to “Serological Response to COVID-19 Vaccination in Saudi Arabia: A Comparative Study of IgG and Neutralising Antibodies Across Vaccine Platforms”
- The current study evaluated IgG and NAb across two dose vaccination platforms in Saudi Arabia. I am concerned about the novelty of this article, as many previous studies published in other countries during the COVID era (2020–2023) have already evaluated antibody responses to both homologous and heterologous vaccination regimens with 2–4 doses and in participants with or without preexisting immunity. Please clarify the benefits highlighted by this study.
- In the abstract mentions “however, lacks a real-world comparative data……”, but when it is read through the main text, it feels like a clinical study. In the methodology, said that to be collected at three time points are from the same or different subjects. It remained unclear.
- please check the number of participants between an abstract/result (1,063 adults) and methodology (Samples collect at three time points: the first samples were collected directly before taking the first dose (Day 0) in a total of 1082 participants, the second samples were collected before taking the 2nd dose, for 277 participants, and the third samples were collected after 20 days or more from the 2nd dose, for 96 participants. The fourth group of participants were known infected COVID-19 cases, although they had taken the vaccine. We collected samples from 66 participants from this group.
- Please correct “The disease was named by the World Health Organisation (WHO) as coronavirus 2019 (COVID-19) (4).” It should be coronavirus disease 2019.
- Please update and correct the information in this sentence “The vaccine is used for active immunisation for individuals 16 years and older.”
- The introduction is missing a review of previously published relevant studies, which mentioned related study design and existing regimens used for COVID-19 vaccination.
- I suggest you construct a study flow diagram containing four group participants to enhance the readability.
- Are there any inclusion and exclusion criteria? Please explain.
- What is the period of sample collection? It looks like you experimented with the first wave of pandemics, and as a result of 5.9% have a history of previous infection.
- Why is infection history unavailable for 969 of 1,063 participants (91.2%)? The authors should consider identifying prior natural infection through the detection of anti-nucleocapsid IgG, which would help explain the phenomenon of hybrid immunity in relation to the vaccination plan. To improve the manuscript, more experiments should be considered.
- The author may consider evaluating the NAb against newer types of Omicron NB lineage; the result will show that the Wu-1 original-based vaccine from Pfizer, Moderna, and Astrazeneca cannot induce NAb against the newer type, as shown in the other published literature.
- In the discussion, the authors should compare the current study with relevant previous research, highlighting similarities and differences, and clearly explaining the novelty of this article.
- According to comment 11, I don’t think the original WU-1-based vaccine is recommended for use nowadays. Many Omicron variants are long-distance from Wuhan. Your discussion should mention the study of the newer COVID-19 vaccine that relates to the WHO’s recommendation. Please update your discussion.
14 Again, I don’t think the vaccinations are necessary for those who have previous immunity by vaccination and hybrid immunity. As the vast majority of the population (>90%) who lived through the pandemic era have experienced SARS-CoV-2 infection, most vaccinated individuals are likely to possess hybrid immunity. This widespread hybrid immunity not only enhances individual protection but also contributes to population-level herd immunity, thereby reducing transmission and the overall burden of disease.
- The WHO continues to recommend completing a full course of COVID-19 vaccination, including with newer vaccine formulations, for individuals who have never been infected or vaccinated.
- The overall discussion should be revised to reflect the current COVID-19 situation, for example, newer vaccines, latest COVID-19 variants, hybrid immunity, herd immunity, and current WHO guidelines regarding updated vaccines and booster strategies, or your country's policy.
- All figures, please consider adding the Unit of measurement in the Y-axis title.
- Figure 5 panel Right-bottom, I feel confused what is the definition of VAC-A to VAC-D, respectively.
Author Response
Reviewer2:
- The current study evaluated IgG and NAb across two dose vaccination platforms in Saudi Arabia. I am concerned about the novelty of this article, as many previous studies published in other countries during the COVID era (2020–2023) have already evaluated antibody responses to both homologous and heterologous vaccination regimens with 2–4 doses and in participants with or without preexisting immunity. Please clarify the benefits highlighted by this study.
Response: We appreciate this important comment. Although we acknowledge that several studies from other countries have evaluated antibody responses to homologous and heterologous vaccination regimens, our study add values in two ways:
1) It provides data specific to the Saudi Arabian population, where the distribution of vaccine types, prior infection rates, and demographic characteristics might differ from those reported elsewhere; local data are critical to inform national public health policies.
2) Our study directly compares IgG and NAb responses across two-dose platforms within this population, thereby offering region-specific insights where published data remain scarce.
We have revised the discussion section and amended to highlight these contributions.
- In the abstract mentions “however, lacks a real-world comparative data……”, but when it is read through the main text, it feels like a clinical study. In the methodology, said that to be collected at three time points are from the same or different subjects. It remained unclear.
Response: Thank you for pointing this out. We have revised the abstract for clarity and to ensure alignment with the existing literature. The sentence now reads: “However, region-specific real word comparative data on their immunogenicity remain limited”.
We also clarify that this was an observational study, as stated in the Methods section of the abstract, in which samples at the predefined time points were collected from the participants.
- please check the number of participants between an abstract/result (1,063 adults) and methodology (Samples collect at three time points: the first samples were collected directly before taking the first dose (Day 0) in a total of 1082 participants, the second samples were collected before taking the 2nd dose, for 277 participants, and the third samples were collected after 20 days or more from the 2nd dose, for 96 participants. The fourth group of participants were known infected COVID-19 cases, although they had taken the vaccine. We collected samples from 66 participants from this group.
Response: Thank you for your comment. The participants numbers have been carefully revised to ensure consistency throughout the manuscript. The updated numbers reflect the dataset after data cleaning, along with adjustments made in response to Reviewer 1’s comments.
- Please correct “The disease was named by the World Health Organisation (WHO) as coronavirus 2019 (COVID-19) (4).” It should be coronavirus disease 2019.
Response: Thank you for pointing this out. We have amended the sentence to read: “The disease was named by the World Health Organization (WHO) as coronavirus disease 2019 (COVID-19)”
- Please update and correct the information in this sentence “The vaccine is used for active immunisation for individuals 16 years and older.”
Response: Thank you for the comment. We have revised the sentence for accuracy and it now reads: “The vaccine is used for active immunization to prevent COVID-19 in individuals 12 years of age and older”.
- The introduction is missing a review of previously published relevant studies, which mentioned related study design and existing regimens used for COVID-19 vaccination.
Response: Thank you for the comment. We have added relevant references to the manuscript’s introduction.
- I suggest you construct a study flow diagram containing four group participants to enhance the readability.
Response: Thank you for your suggestion. We have added a flow chart as suggested.
- Are there any inclusion and exclusion criteria? Please explain.
Response: We have added further clarification regarding inclusion & exclusion criteria in the methods section. The revised text now reads as follows:
“The study includes Saudis and non-Saudis, healthy males and females aged 18 years and older vaccinated with one of the following vaccines: Moderna (mRNA-1273), Johnson & Johnson (JNJ-78436735), BNT162b2 Pfizer BioNTech, and ChAdOx1 nCoV-19 (AstraZeneca), excluding pregnant women and individuals with previous viral infection. Samples were collected at three time points: Baseline, Post-dose 1, Post-dose 2”
- What is the period of sample collection? It looks like you experimented with the first wave of pandemics, and as a result of 5.9% have a history of previous infection.
Response: We have revised this point and added the sample collection period. The text now reads as follow: “The samples were collected from participants visiting the vaccination sites in the Riyadh region in the period between October 2021 and July 2022.”
- Why is infection history unavailable for 969 of 1,063 participants (91.2%)? The authors should consider identifying prior natural infection through the detection of anti-nucleocapsid IgG, which would help explain the phenomenon of hybrid immunity in relation to the vaccination plan. To improve the manuscript, more experiments should be considered.
Response: Thank you for this valuable suggestion. We agree that assessing prior infection through anti-N IgG would provide further insights into hybrid immunity. However, due to limited resources, we were unable to perform additional experiments.
- The author may consider evaluating the NAb against newer types of Omicron NB lineage; the result will show that the Wu-1 original-based vaccine from Pfizer, Moderna, and Astrazeneca cannot induce NAb against the newer type, as shown in the other published literature.
Response: Thank you for this suggestion. We agree that evaluating NAbs against newer Omicron NB-lineage subvariants is scientifically important. However, this is beyond the scope of the present study, which focused on comparative responses to primary two-doses vaccination platforms available at the time of data collection. In addition, it has already been well documents in the literature that first generation Wu-1-based vaccines elicit little or no neutralizing activity against these newer Omicron lineages. To highlight this point, we have added a statement in the Discussion acknowledging this limitation and cited the relevant studies. More specifically, we included the following:
“This limitation is particularly relevant given published evidence that first-generation Wu-1-based vaccines elicit little or no neutralizing activity against newer Omicron NB sublineages, underscoring the importance of updated vaccine formulations (29–31) Recent studies have shown that bivalent and updated mRNA vaccines targeting Omicron sub-lineages elicit broader neutralization profiles, reflecting the continual evolution of vaccine strategies in response to emerging variants (30,31).”
- In the discussion, the authors should compare the current study with relevant previous research, highlighting similarities and differences, and clearly explaining the novelty of this article.
Response: Thank you for your comment. We have expanded the Discussion as per suggestion.
The following statements have been added and highlighted in the revised manuscript:
- “ Reinforces the likelihood that undocumented previous infections contributed to our cohort’s relatively high seroprevalence”
- “ Our results align with recent international reports, such as Barros-Martins et al. (2021) and Pozzetto et al. (2021) (10,21), that demonstrated that heterologous vaccine regimens, especially combinations of vector-priming and mRNA boosting, appear to enhance immunogenicity”
- “Similarly, durability of IgG responses has been observed in regional studies from Saudi Arabia (22). However, unlike those studies, our work directly assessed both binding (IgG) and functional neutralizing antibodies across multiple two-dose platforms within the Saudi population. This provides novel, region-specific evidence that complements the global literature and addresses the limited data available from the Middle East.”
- “This concordance also adds confidence that our serological findings are meaningful proxies of protective immunity, as highlighted in multiple reviews (24,25).”
- “Further Saudi-based age-stratified studies are needed to explore the impact of these variables, particularly in the elderly and immunocompromised populations.”
- “This novelty strengthens the contribution of our work, since regional data on COVID-19 vaccine immunogenicity remain sparse compared to Western cohorts yet are essential for shaping public health policies tailored to Saudi Arabia.”
- “This limitation is particularly relevant given published evidence that first-generation Wu-1-based vaccines elicit little or no neutralising activity against newer Omicron NB sublineages, underscoring the importance of updated vaccine formulations (29–31) Recent studies have shown that bivalent and updated mRNA vaccines targeting Omicron sublineages elicit broader neutralization profiles, reflecting the continual evolution of vaccine strategies in response to emerging variants (30,31).”
- “This reinforces the importance of building national immunological surveillance programs, as recently emphasized in Saudi Vision 2030 health security initiatives.”
- “Taken together, our findings not only confirm global trends but also provide much-needed regional data that can inform Saudi vaccination policy.”
- According to comment 11, I don’t think the original WU-1-based vaccine is recommended for use nowadays. Many Omicron variants are long-distance from Wuhan. Your discussion should mention the study of the newer COVID-19 vaccine that relates to the WHO’s recommendation. Please update your discussion.
Response: Thank you for this important point. We agree that first-generation Wu-1-based vaccines are no longer recommended as primary vaccination due to substantial antigenic distance between Wuhan-Hu-1 and currently circulating Omicron lineages. We have amended the discussion to reflect this point and included WHO recommendations supporting the use of updated, Omicron-adapted vaccines and clarified that our study focused on the earlier vaccination phase in Saudi Arabia.
- Again, I don’t think the vaccinations are necessary for those who have previous immunity by vaccination and hybrid immunity. As the vast majority of the population (>90%) who lived through the pandemic era have experienced SARS-CoV-2 infection, most vaccinated individuals are likely to possess hybrid immunity. This widespread hybrid immunity not only enhances individual protection but also contributes to population-level herd immunity, thereby reducing transmission and the overall burden of disease.
Response: Thank you for raising this point. We agree that hybrid immunity plays an important role in shaping population-level protection against COVID-19, and that many vaccinated individuals are likely to have acquired some degree of hybrid immunity. We have added a statement in the Discussion to acknowledge this important context. However, we would like to clarify that the objective of our study was not to evaluate vaccine policy in the setting of hybrid immunity, but rather to assess and compare the immunogenicity of primary two-dose vaccine regimens during the initial vaccination rollout in Saudi Arabia.
- The WHO continues to recommend completing a full course of COVID-19 vaccination, including with newer vaccine formulations, for individuals who have never been infected or vaccinated.
Response: Thank you for your comment. We have noted in the Discussion that WHO recommends a full course of updated COVID-19 vaccination for individuals without prior infection or vaccination, while clarifying that our study focused on first-generation vaccines during the initial rollout in Saudi Arabia.
- The overall discussion should be revised to reflect the current COVID-19 situation, for example, newer vaccines, latest COVID-19 variants, hybrid immunity, herd immunity, and current WHO guidelines regarding updated vaccines and booster strategies, or your country's policy.
Response: Thank you for this valuable suggestion. We have revised the Discussion to reflect the current COVID-19 situation. We also added a brief reference to Saudi Arabia’s vaccination policy to place our findings in context.
- All figures, please consider adding the Unit of measurement in the Y-axis title.
Response: Thank you for your comment. We have considered that and updated the figures accordingly.
- Figure 5 panel Right-bottom, I feel confused about the definition of VAC-A to VAC-D, respectively.
Response: Thank you for your comment. We have renumbered and amended all figures to improve clarity and avoid confusion.

Round 2
Reviewer 1 Report
Comments and Suggestions for Authors
Most of the issues were resolved according to the reviewers' comments
Some minor issues are recommended to be resolved prior to the manuscript might be recommended for publication
1) Please remove "Boxplot" from Figures' captions
2) Please rename Table 2 according to the results presented
3) Please check the typos (impired) line 163-164
4) Please, indicate on the boxplots, if any of boxes have statistically significant differences
Sincerely,
Author Response
Reviewer comments: We greatly appreciate your insightful comments, which have contributed to improving the quality and clarity of the paper.
- Comment: Please remove "Boxplot" from Figures' captions
Response: The term “Boxplot” has been removed from all figure captions, as requested.
2- Comment: Please rename Table 2 according to the results presented
Response: Table 2 has been renamed to accurately reflect the results presented.
- Comment: Please check the typos (impired) line 163-164
Response: The typo (impired) in lines 163–164 has been corrected.
- Comment: Please, indicate on the boxplots if any of the boxes have statistically significant differences
Response: Statistical significance has been clearly indicated on the boxplots where applicable.
All revisions have been incorporated into the manuscript, and the changes are highlighted using track changes for ease of review.
Reviewer 2 Report
Comments and Suggestions for Authors
After revision, I would accept for publication.
Author Response
We greatly appreciate your insightful comments, which have contributed to improving the quality and clarity of the paper.